# Predictors of Workplace Substance Reuse among Patients with Alcohol or Illegal Substance Use Disorder in the Workplace

**DOI:** 10.3390/ijerph191610023

**Published:** 2022-08-14

**Authors:** Su-Ting Hsu, Hung-Chi Wu, Hui-Tzu Chien, Dian-Jeng Li

**Affiliations:** 1Department of Community Psychiatry, Kaohsiung Municipal Kai-Syuan Psychiatric Hospital, Kaohsiung 80276, Taiwan; 2Department of Medicine, Kaohsiung Medical University Hospital, Kaohsiung 80708, Taiwan; 3Department of Addiction Science, Kaohsiung Municipal Kai-Syuan Psychiatric Hospital, Kaohsiung 80276, Taiwan; 4Department of Nursing, Shu-Zen Junior College of Medicine and Management, Kaohsiung 82144, Taiwan; 5Department of Nursing, Meiho University, Pingtung 91200, Taiwan

**Keywords:** substance use, alcohol use, workplace, predictor, reuse

## Abstract

Substance and alcohol use in the workplace have become a global health burden; however, the etiologies have seldom been explored. The aims of this study were to develop a Workplace Substance Reuse Questionnaire (WSRQ) to measure the multidimensional factors associated with the reuse of alcohol or illegal substances in the workplace. The predictors of reuse were also investigated. The WSRQs for alcohol (WSRQ-Alc) and illegal substances (WSRQ-Sub) were composed of 15 and 13 items, respectively. Factors associated with workplace substance reuse included workplace environment, workload, social interaction in the workplace and other cues. Construct validity and reliability were performed to verify the questionnaires. Multivariate linear regression was conducted to estimate the associations between the factors and WSRQ score. A total of 90 patients with substance or alcohol use disorder were recruited. The results demonstrated that the WSRQ-Alc and WSRQ-Sub had acceptable reliability, with variance of 76.4% and 75.4%, respectively. The confirmatory factor analysis fit indices also indicated the adequacy of the model. A longer duration of alcohol use (β = 0.44; *p* = 0.002) and higher frequencies of changing job (β = 0.32; *p* = 0.027) and working part time (β = 0.32; *p* = 0.028) were significantly associated with higher WSRQ-Alc score. Our results highlight the importance of abstinence treatment and job referral for individuals with alcohol or substance use. Further studies are warranted to help extend the applicability and generalizability of the WSRQ.

## 1. Introduction

Alcohol and substance use has become a serious global concern. A national epidemiological survey in the US reported that alcohol use disorder (AUD) is a highly prevalent, comorbid, disabling disorder that often goes untreated, with a lifetime prevalence of around 29% [1]. In a global study of alcohol, tobacco and illicit drug use, the estimated prevalence in adults was 18.4% for heavy episodic alcohol use in the most recent month and 3.8%, 0.77%, 0.37% and 0.35% for cannabis, amphetamine, opioid and cocaine use in the past year, respectively [2]. The workplace has been shown to play an important role in the problem of substance abuse in the work force and multidimensional interventions of prevention, detection and control of alcohol and other drugs have been applied [3]. In addition, most users of illicit drugs and heavy drinkers have been shown to be working adults [3]. Moreover, the workplace consequences of problematic substance use include impaired work functioning, work-related stress and anxiety, bullying and traumatic clinical incidents [4]. Another study also exhibited the importance of working environment in smoking, manifesting the efficacy of the Tobacco-free Workplace Program [5]. As alcohol and substance use in the workplace seriously affect productivity, it is important to identify the etiologies of their use in the workplace.

When investigating the complicated etiologies of substance and alcohol use, it is crucial to explore the factors affecting their reuse or relapse. These factors may be multidimensional and a previous study reported that psychiatric co-morbidity, craving, use of other substances, health and social factors were consistently significantly associated with relapse of alcohol use disorder [6]. Another study indicated that relapse of substance use was associated with status of substance use in family members and shorter duration of treatment with substance-related problems [7]. In addition, being male and unemployed were associated with a higher relapse rate of substance use disorder (SUD) in military veterans [8]. Furthermore, psychosocial and environmental factors are also strongly associated with relapse, such as drug availability, family disputes, an addicted friend, close addicted relatives, being homeless and living in single-parent families [8,9,10]. For individuals with alcohol use problems, divorced status, longer duration of alcohol dependence, younger age at initiation of alcohol use, younger age at initial admission due to alcohol-related problems and comorbidity with other mental illnesses have been associated with an increased risk of relapse of alcohol use [11].

The predictors associated with relapse of substance or alcohol use have been investigated. However, it is also crucial to investigate the specific predictors of relapse in the workplace due to the following reasons: (1) most illegal substance users and heavy drinkers are members of the workforce [3]; (2) substance and alcohol abuse have a negative impact on the health of workers, safety and productivity; and (3) the workplace is an important area for social networking, which may influence the pattern of substance or alcohol use. Moreover, comprehensive tools to assess workplace-specific factors, such as workplace environment, satisfaction of the job or workplace and social interaction in the workplace, are lacking, where some of these factors may contribute to the relapse of substance or alcohol use in the workplace. In addition, predictors associated with relapse of substance or alcohol use in the workplace are still unexplored. Given the gaps in the knowledge, the aim of this study was to develop useful questionnaires to effectively measure the multidimensional factors associated with the reuse of alcohol or illegal substances in the workplace. The reliability and validity of the questionnaires were also estimated. We hypothesized that we could develop a useful tool to identify the environmental, social and other factors associated with relapse of substance or alcohol use. Several factors may also be identified to predict relapse of substance or alcohol use in the workplace.

## 2. Methods

### 2.1. Participants, Procedures and Ethics

Participants were recruited from the outpatient department of addiction treatment at Kaohsiung Municipal Kai-Syuan Psychiatric Hospital (KSPH). Printed advertisements were posted in the public area of the hospital to recruit participants. The recruitment period was from 12 November 2019 to 28 November 2020. We designed the current study as a cross-sectional survey and paper-and-pencil questionnaires were used. Research assistants individually explained the procedures to the participants and helped them complete the questionnaires. The inclusion criteria were participants who: (1) were diagnosed with AUD or SUD according to DSM-5; (2) were followed up at the outpatient department of addiction treatment in our hospital; (3) could understand the objectives of the current study and follow the instructions; (4) were aged at least 18 years; and (5) gave informed consent prior to participation. Participants with missing data and those who could not complete the questionnaires were excluded. The sample size was determined using G-Power software [12]. The method of multi-variate linear regression was applied and the alpha value was set at 0.05, with a power of 0.80. We set the effect size at a medium effect [13] due to insufficiency of previous studies. The minimum required sample size was 74 patients. Considering the possibility of dropout, the required sample size was preliminarily set as 90 subjects. The current study was approved by the Institutional Review Board of KSPH (approval no. KSPH-2019-13) and was conducted according to the current revision of the Declaration of Helsinki.

### 2.2. Measure

#### 2.2.1. Workplace Substance Reuse Questionnaire

The Workplace Substance Reuse Questionnaire (WSRQ) was constructed to estimate the associations of workplace-related factors with the desire to reuse a substance. To verify the face validity of the WSRQ, we held expert meetings to review the items of the adequacy of the questionnaires and remove irrelevant items. Five experts and translators were invited to verify the face and content validity.

The original WSRQ was composed of 20 items. We removed some items to improve the construct validity and created two versions of the questionnaire. The WSRQ for alcohol use (WSRQ-Alc) contained 15 items encompassing four factors: workplace environment, workload, social interaction in the workplace and other cues for drinking. The WSRQ for illegal substance use (WSRQ-Sub) contained 13 items and encompassed the same four factors as the WSRQ-Alc. Workplace environment included questions on undesirable elements of the environment in the workplace, such as sultry air or loud noise. Workload included questions on intolerable workload, such as working overtime or irregular shifts. Social interaction included questions on maladaptive interactions with colleagues, directors or bosses. Other cues included questions on other unclassified factors. All of the above factors were discussed and formulated in the expert meetings. The questions were scored using a 2-point Likert scale, as 0 (no/never) and 1 (yes/ever). A higher total score on the WSRQ indicated a higher level of risk in substance reuse in the workplace. The WSRQ-Alc and WSRQ-Sub are provided in Appendix A.

#### 2.2.2. Demographic Characteristics

Demographic characteristics were recorded including age, sex, duration of alcohol or illegal substance use, job status (full time or part time), monthly income (NTD ≤ 23,100 or >23,100), professional license (do not have or have), education (high school/below or college/above), job changes since initiating substance use (never, less than twice, three to five times, more than five times) and unemployed months in the recent one year. The monthly minimum income threshold was set at NTD 23,100 for a full-time job in Taiwan. Professional license was defined as a license related to the participant’s career. We recorded job changes and unemployed months to identify job stability among the participants.

### 2.3. Statistical Analysis

All statistical analyses were performed using SPSS version 23.0 for Windows (IBM, Armonk, NY, USA). Descriptive analysis was used to summarize the variables, including age, sex, duration of alcohol or illegal substance use, job status, income, professional license, education, job changes since initiating substance use and unemployed months in the recent one year. Pearson’s χ^2^ test and analysis of variance (ANOVA) were used to compare differences in the demographic characteristics between groups of only alcohol use, only illegal substance use and both substance and alcohol use. The duration of alcohol or illegal substance was independently reported because it could only be divided into two groups. In order to test the reliability of the questionnaires in the current study, internal consistency was estimated with Cronbach’s α, where a value >0.6 indicated moderate reliability [14]. Exploratory factor analysis (EFA) and confirmatory factor analysis (CFA) were applied to estimate construct validity. The EFA was estimated using SPSS version 23.0 for Windows (IBM, Armonk, NY, USA). Initially, the Kaiser–Mayer–Olkin (KMO) measure of sampling adequacy and Bartlett test were used to examine the adequacy of the EFA. The data were acceptable for factor analysis if the KMO value was >0.60 and significance (*p* < 0.05) was identified in the Bartlett test [15]. According to the assumption that the factors were associated, principal axis factor analysis was used with Varimax rotation. Total variance explained (%) and factor loadings were also identified. Variance indicates how well a relevant notion can be measured [16] and an acceptable threshold of total variance explained is at least 60% [17].

The CFA was conducted using Amos statistical software (IBM Amos Statistics for Windows, Version 23.0, IBM, Armonk, NY, USA). Since significance in the Kolmogorov–Smirnov test (*p* < 0.001) stood for non-normal distribution for the CFA subsample, maximum likelihood with Satorra–Bentler correction was applied to identify if the model fit with the structure of the item factor. Average variance extracted (AVE) and composite reliability (CR) were applied to estimate the discriminant validity and convergent validity. CR and AVE levels of 0.6 and 0.33, respectively, were reported to indicate satisfactory validity [18]. In order to test the adequacy of the model with CFA, multiple indices were applied to verify the goodness of fit. For each index, the values indicating requirable model fit were as follows: incremental fit index (IFI ≥ 0.9); chi-square goodness-of-fit test (χ^2^/df < 5.0); Tucker–Lewis index (TLI ≥ 0.9); comparative fit index (CFI ≥ 0.9); and root-mean square error of approximation (RMSEA < 0.08) [19].

Finally, we further tested the associations between the level of WSRQ and related factors. We transformed the categorical variable of job changes into a continuous variable (never = 1, less than twice = 2, three to five times = 3, more than five times = 4) because the items of this question were also continuous. Since the dependent outcome (WSRQ-Alc or WSRQ-Sub) was a continuous variable, multivariate linear regression was conducted to ascertain the independent factors with the alpha level set at 0.05.

## 3. Results

### 3.1. Summary of Demographic Analysis

In total, 90 subjects completed the questionnaires and were entered in the analysis, including 23 with only alcohol use, 42 with only illegal substance use and 25 with both alcohol and illegal substance use. The average duration of alcohol use was 14.37 ± 9.91 years and the average duration of illegal substance use was 5.42 ± 6.7 years. There were no significant differences between groups in continuous and categorical variables. Details of the characteristics for all participants are listed in Table 1.

### 3.2. Reliability Test and Exploratory Factor Analysis for Construct Validity

The overall internal consistency coefficient (Cronbach’s α) of the WSRQ-Alc was 0.93 and the value of each factor was within 0.83 to 0.9, indicating adequate reliability. The overall Cronbach’s α of the WSRQ-Sub was 0.89,and the value of each factor was around 0.75 to 0.93. Details are shown in Table 2 and Table 3.

For scales of the WSRQ-Alc, the KMO value of sampling adequacy was 0.748, which was within the requirable range. Bartlett’s test of sphericity, which assesses whether a matrix differentiates from the identity matrix, demonstrated a significant result (*p* < 0.001). This demonstrated that the WSRQ-Alc was suitable for factor analysis. In addition, the WSRQ-Alc explained 76.39% of the total variance, which was within an acceptable range. Similarly, the KMO value was 0.82 for the WSRQ-Sub and Bartlett’s test revealed a significant result (*p* < 0.001). The WSRQ-Sub explained 75.4% of the total variance, which was within a requirable range. In short, the results of EFA showed that each factor extracted from every item could interpret all items on the WSRQ-Alc and WSRQ-Sub appropriately. The details of EFA are listed in Table 2 and Table 3.

### 3.3. Confirmatory Factor Analysis for Construct Validity

After performing CFA, the CR value of each factor was 0.83 to 0.902 and the AVE for each factor was 0.612 to 0.709 for the WSRQ-Alc. All of the values from the validation test were within an acceptable range, indicating good convergent and discriminant validity [15]. The fit indices for the WSRQ-Alc indicated that the indices fitted or were close to adequacy of the model (CFI = 0.84; IFI = 0.85; TLI = 0.82; χ^2^/df = 1.956; SRMR = 0.06). CFA of the WSRQ-Sub demonstrated CR values of 0.755 to 0.934 and AVE values of 0.442 to 0.823, indicating that the convergent and discriminant validity were also acceptable. The adequacy of the CFA model also fitted the threshold of requirement (CFI = 0.93; IFI = 0.93; TLI = 0.91; χ^2^/df = 1.576; SRMR = 0.08). Overall, the WSRQ showed good validity, ensuring that the operational definition was comparable to the conceptual definition. Details of the CFA are listed in Appendix A.

### 3.4. Predictors of the WSRQ

The results of the multivariate linear regression revealed that a longer duration of alcohol use (β = 0.44; *p* = 0.002), more frequently changing job (β = 0.32; *p* = 0.027) and working part time (β = 0.32; *p* = 0.028) were significantly associated with a higher WSRQ-Alc score (Table 4). On the other hand, more frequently changing job (β = 0.21; *p* = 0.092) demonstrated a non-significant trend of an association with a higher WSRQ-Sub score.

## 4. Discussion

### 4.1. Main Findings of the Current Study

In the current study, we developed the WSRQ and tested its reliability and validity. The construct validity and reliability supported the adequacy of the scale’s psychometric properties, confirming that the WSRQ is a brief and feasible tool to estimate the level of risk of substance reuse in the workplace in a multiple-dimensional approach. Moreover, we also found that subjects with a longer duration of alcohol use, those who changed job more frequently after initiating substance use and those who worked part-time jobs were significantly associated with a higher WSRQ-Alc score, indicating a higher risk of alcohol use in the workplace.

### 4.2. Multidimensional Factors in the Workplace

Circumstances related to the workplace, such as workload and workplace environment, comprised the major parts of the WSRQ. Shaw et al. reported that the association between substance use and work-related factors may be mediated by work-related pain or discomfort [20]. Low satisfaction with the workplace, keyboard position close to the body, low work task variation and self-perceived medium/high muscular tension were found to be risk factors for the development of neck pain [21]. Another study also showed an association between workplace injury and opioid misuse and it elaborated the important role of occupational health nurses in the prevention of substance use in the workplace [22]. The WSRQ estimates not only the impact of heavy work but also physical environmental factors, such as noise, smells and air quality. The physical environment has seldom been investigated as a risk factor for substance reuse and the WSRQ provides a novel approach to evaluate this risk. However, further studies may be needed to verify its application.

Another important factor explored by the WSRQ is problems in social interactions between workers and their colleagues, directors or bosses. These maladaptive interactions may result in a great psychosocial burden in the workplace. Psychosocial job stressors, such as high job demands, burn out and low skill discretion, have been associated with a higher risk of substance abuse [23,24]. Another study also found that opioid use disorder was associated with low skill discretion, high psychological burden and job strain [23]. Therefore, the negative impact of maladaptive social interactions in the workplace may also be an important dimension when estimating the risk of substance or alcohol use. In addition, we also included factors associated with self-efficacy, such as “achievement”, “satisfaction” and “nothing to do”. Self-efficacy has been negatively associated with alcohol and substance use among students [25]. Another study demonstrated that higher frequency of alcohol use was associated with lower self-efficacy [26]. The WSRQ-Sub also asks about invitations from colleagues to use a substance. Since friends’ substance use has been shown to be a predictor of substance use [27], it is reasonable to estimate the impact of colleagues on substance use in the workplace.

### 4.3. Predictors of Alcohol or Illegal Substance Reuse in the Workplace

Among the predictors of alcohol reuse in the workplace, a longer duration of alcohol use predicted a higher WSRQ-Alc score. The duration of alcohol use has been shown to predict both relapse of drinking and also greater alcohol consumption [11,28]. Moreover, a longer duration of alcohol use has been associated with morphological and biological impairment of the brain [29] and inflammatory system [30,31]. Consistent with previous studies, we identified the impact of duration of alcohol use on the reuse of alcohol in the workplace. We also found that reuse of alcohol in the workplace was associated with job stability, such as frequently changing job and part-time work. In addition, changing job was insignificantly associated with substance reuse in the workplace. Previous studies reported associations between injecting opioids in public, daily opioid injections and unstable housing with irregular work and illegal employment [32]. Furthermore, a previous study recruiting patients with substance use disorder demonstrated that most participants (59%) had relatively long histories of unemployment and underemployment, while the therapeutic workplace intervention was beneficial in stabilizing their jobs [33]. Consistent with these studies, we also found an association between job stability and alcohol use in the workplace.

### 4.4. Limitations

There are several limitations to this study. First, we did not estimate the severity of dependence or craving symptoms, which may have confounded the scores of the WSRQ. Second, the cross-sectional survey design limits the interpretation of the association between factors and WSRQ scores. Finally, a single-center study may limit the generalizability and applicability of the results to other populations.

## 5. Conclusions

We developed the WSRQ and found it to be a valuable and reliable tool to estimate the risk of substance or alcohol use in the workplace with multidimensional factors. Using several categories to assess the factors associated with substance or alcohol use in the workplace, this study establishes a foundation for further research on workplace environment, workload, social interaction in the workplace and self-efficacy. The WSRQ could be used in employee-assistance programs in companies and factories to help prevent and treat alcohol and substance abuse, although further validation is required.

We also identified that patients with a longer duration of alcohol use and job instability were significantly associated with the risk of alcohol use in the workplace. The clinical implication of the current study is that we highlight the importance of timely assessments and interventions, such as abstinence treatment, for individuals with alcohol or substance use in the workplace. With the heavy impact of environmental factors in the workplace on job stability, our results also suggest that multidimensional environmental enrichment interventions (involving physical activity, social interaction, vocational training, recreational and community involvement) may have therapeutic effects for substance or alcohol use disorder [34]. To address the risk of unstable employment, early engagement and further referral to vocational training programs to acquire skills or pre-hiring job matching may be beneficial for individuals with alcohol or substance use. Further studies are warranted to help extend the applicability and generalizability of the present study. For example, a study with longitudinal follow-up and advanced assessment could be helpful to better understand the etiologies of alcohol or substance use in the workplace.

## Figures and Tables

**Table 1 ijerph-19-10023-t001:** Sociodemographic characteristics of participants (n = 90).

Variable	Groups			
	Alcohol	Illegal Substance	Combined	Statistics
Categorical	n (%)	n (%)	n (%)	P
Total	23 (100)	42 (100)	25 (100)	-
Sex				0.390 ^a^
Female	22 (95.7)	39 (92.9)	25 (100)	
Male	1 (4.3)	3 (7.1)	0 (100)	
Job status				0.429 ^a^
Full time	17 (73.9)	35 (83.3)	22 (88)	
Part time	6 (26.1)	7 (16.7)	3 (12)	
Income				0.460 ^a^
≤23,100 (NTD)	6 (26.1)	8 (19)	3 (12)	
>23,100 (NTD)	17 (73.9)	34 (81)	22 (88)	
Professional license				0.504 ^a^
No	11 (47.8)	26 (61.9)	13 (52)	
Yes	12 (52.2)	16 (38.1)	12 (48)	
Education				0.873 ^a^
High school or below	16 (69.6)	30 (71.4)	19 (76)	
College or above	7 (30.4)	12 (28.6)	6 (24)	
Job changes ^b^				0.903 ^a^
Never	9 (39.1)	14 (33.3)	11 (44)	
Less than twice	6 (26.1)	10 (23.8)	5 (20)	
Three to five	4 (17.4)	8 (19)	6 (24)	
More than five	4 (17.4)	10 (23.8)	3 (12)	
Continuous	Mean (SD)	Mean (SD)	Mean (SD)	F ^d^
Age	43 (7.69)	40.81 (9.89)	39.56 (7.09)	0.97 (N.S.)
Jobless months ^c^	2.89 (4.15)	3.36 (4.33)	2.2 (3.11)	0.66 (N.S.)

^a^: Pearson’s χ^2^ test; ^b^: Change job after initiation of drinking or illegal substance use; SD: Standard deviation; ^c^: in recent one year; ^d^: F statistics of analysis of variance; N.S.: Non-statistical significance.

**Table 2 ijerph-19-10023-t002:** Exploratory factor analysis for Workplace Substance Reuse Questionnaire (Alcohol).

Components/Items	EFA (Varimax Rotation) *			Reliability	Factor Loading
	Sum of Squared Loading (Eigenvalue)	Variance Explained (%)	Cumulative Variance Explained (%)	Cronbach’s Alpha	
Workplace environment	3.279	21.861	21.861	0.902	
Dimly illuminated					0.807
Sultry air					0.822
Strong smell					0.896
Loud noise					0.821
Workload	3.255	21.701	43.562	0.885	
Often requires irregular shifts					0.670
Often work overtime					0.731
Excessive physical fatigue without adequate rest					0.749
Mental stress, often needing to be refreshed or lessened					0.731
High repetition of work; no sense of achievement					0.756
Social interaction	3.063	20.421	63.983	0.882	
Frequently occupational hazards					0.680
Bad times with coworkers or bosses					0.841
Bosses or coworkers treat me makes me feel like I’m useless					0.848
Poor atmosphere among colleagues					0.820
Other cues	1.860	12.402	76.385	0.830	
I am not satisfied with the work					0.810
I feel that I have to do this job in order to earn living					0.851

* Kaiser–Meyer–Olkin (KMO) Measure of Sampling Adequacy: 0.748, Bartlett’s Test of Sphericity: <0.001, Overall Cronbach’s Alpha: 0.933.

**Table 3 ijerph-19-10023-t003:** Exploratory factor analysis for Workplace Substance Reuse Questionnaire (Illegal substance).

Components/Items	EFA (Varimax Rotation) *			Reliability	Factor Loading
	Sum of Squared Loading (Eigenvalue)	Variance Explained (%)	Cumulative Variance Explained (%)	Cronbach’s Alpha	
Social interaction	2.695	20.734	20.734	0.925	
Bad times with coworkers or bosses					0.831
Bosses or coworkers treat me makes me feel like I’m useless					0.874
Poor atmosphere among colleagues					0.786
Workplace environment	2.471	19.007	39.741	0.837	
Dimly illuminated					0.867
Strong smell					0.834
Loud noise					0.704
Workload	2.403	18.481	58.222	0.805	
Often requires irregular shifts					0.755
Often work overtime					0.802
Excessive physical fatigue without adequate rest					0.675
Other cues	2.234	17.181	75.404	0.748	
High repetition of work; no sense of achievement					0.759
Sitting there for a long time with nothing to do					0.694
Colleagues will invite me to use					0.670
The company do not claim that substance is not allowed					0.632

* Kaiser–Meyer–Olkin (KMO) Measure of Sampling Adequacy: 0.820, Bartlett’s Test of Sphericity: <0.001, Overall Cronbach’s Alpha: 0.893.

**Table 4 ijerph-19-10023-t004:** Predictors of Workplace Substance Reuse Questionnaire examined by multivariate linear regression.

Predictors	WSRQ-Alc				WSRQ-Sub			
Continuous variables	β	t	95% CI	p	β	t	95% CI	p
Age (years)	0.09	0.58	−0.14, 0.26	0.566	−0.05	−0.42	−0.14, 0.09	0.680
Duration of use ^a^	0.44	3.28	0.09, 0.35	**0.002**	−0.06	−0.46	−0.19, 0.12	0.647
Job changes ^b^	0.32	2.28	0.18, 2.82	**0.027**	0.21	1.71	−0.13, 1.64	0.092
Jobless months in recent one year	0.08	0.53	−0.28, 0.48	0.599	−0.18	−1.45	−0.45, 0.07	0.152
Categorical variables					β	t	95% CI	p
Education level								
High school or below	Ref	−	**−**	−	Ref	−	**−**	−
College or above	−1.71	−1.78	−5.16, 1.35	0.245	0.09	0.73	−1.47, 3.16	0.470
Sex								
Male	Ref	−	**−**	−	Ref	−	**−**	−
Female	−0.05	−0.31	−11.84, 8.69	0.759	0.12	0.98	−2.50, 7.39	0.326
Income								
≦23,100 (NTD)	Ref	−	**−**	−	Ref	−	**−**	−
>23,100 (NTD)	−0.26	−1.79	−6.6, 0.38	0.08	−0.03	−0.27	−3.15, 2.41	0.792
Professional license								
No	Ref	−	**−**	−	Ref	−	**−**	−
Yes	0.24	1.71	−0.43, 5.28	0.094	0.11	0.87	−1.17, 2.99	0.386
Comorbidity ^c^								
No	Ref	−	**−**	−	Ref	−	**−**	−
Yes	−0.25	−1.75	−5.31, 0.38	0.088	−0.05	−0.43	−2.59, 1.67	0.667
Job status								
Full time	Ref	−	**−**	*−*	Ref	−	**−**	−
Part time	0.32	2.27	0.44, 7.29	**0.028**	0.15	1.21	−1.13, 4.59	0.231

^a^: Duration of drinking for WSRQ-Alc and duration of illegal substance use for WSRQ-Sub; ^b^: Change job after initiation of drinking or illegal substance use; ^c^: Comorbid with illegal substance use for WSRQ-Alc and comorbidity with alcohol use for WSRQ-Sub; WSRQ-Alc: Workplace Substance Use Questionnaire for alcohol use; WSRQ-Sub: Workplace Substance Use Questionnaire for illegal substance use; β: standardized coefficients; t: T score; CI: Confidence interval; SD: Standard deviation; **Bolds:***p* < 0.005.

## Data Availability

The data presented in this study are available on request from the corresponding author.

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
