# Peer review of "Predictors of Workplace Substance Reuse among Patients with Alcohol or Illegal Substance Use Disorder in the Workplace"

_ijerph, 2022, doi:10.3390/ijerph191610023_

Round 1
Reviewer 1 Report
Dear Authors congratulations that you have created a questionnaire and validated it and successfully completed your research.
I am not quite sure about the format of this journal but please read the instructions to the author on the journal website. In my opinion, you do not need to put many sub titles in your introduction.
Your method session should start with study design and you must mention whether you have calculated the sample size? What is your calculated minimum sample size? How did you do your sampling ( is it convenient sampling or purposive sampling or random sampling?).
I am impressed that you have used multivariate linear regression. I will jot give comment on that as I am not that expert in it.
I felt that your conclusion need to be concised.
Author Response
as attached file

Reviewer 2 Report
The topic of this research is important. We congratulate the authors for such an interesting study. however, relevant improvements need to be done to their manuscript.
1. The introduction needs to be improved. authors need to highlight the importance of their research, the gap they found, the main previous research as well as a brief literature review.
2. hypotheses of the research need to be included and theoretical support for them needs to be developed.
3. authors only include 27 references. of those 27, only 9 are of the last five years. authors need to make an extensive literature review and include at least 60% of their references of the last five years.
4. references need to include DOI.
5. profesional implications need to be included as well as recommendations.
Author Response
as attached file

Reviewer 3 Report
Dear author, having read your work, I congratulate you on it.
I recommend you to update your bibliography as less than half of it is less than 5 years old, which may negatively influence the presentation of your results.
Author Response
as attached file

Round 2
Reviewer 2 Report
the authors have made the corrections needed, I recommend to accept the manuscript in present form.